# Colossal negative magnetoresistance in field-induced Weyl semimetal of magnetic half-Heusler compound

Kentaro Ueda [1] ✉, Tonghua Yu [1], Motoaki Hirayama[1,2], Ryo Kurokawa[1], Taro Nakajima [2,3], Hiraku Saito[3], Markus Kriener [2], Manabu Hoshino [2], Daisuke Hashizume [2], Taka-hisa Arima [2,4], Ryotaro Arita [2,5] & Yoshinori Tokura [1,2,6]

The discovery of topological insulators and semimetals triggered enormous interest in exploring emergent electromagnetic responses in solids. Particular attention has been focused on ternary half-Heusler compounds, whose electronic structure bears analogy to the topological zinc-blende compounds while also including magnetic rare-earth ions coupled to conduction electrons. However, most of the research in this system has been in band-inverted zero-gap semiconductors such as GdPtBi, which still does not fully exhaust the large potential of this material class. Here, we report a less-studied member of half-Heusler compounds, HoAuSn, which we show is a trivial semimetal or narrow-gap semiconductor at zero magnetic field but undergoes a field-induced transition to a Weyl semimetal, with a negative magnetoresistance exceeding four orders of magnitude at low temperatures. The combined study of Shubnikov-de Haas oscillations and first-principles calculation suggests that the exchange field from Ho 4$f$ moments reconstructs the band structure to induce Weyl points which play a key role in the strong suppression of large-angle carrier scattering. Our findings demonstrate the unique mechanism of colossal negative magnetoresistance and provide pathways towards realizing topological electronic states in a large class of magnetic half-Heusler compounds.

A Weyl semimetal is characterized by pairs of linearly band-crossing points (Weyl points) with nondegenerate bands in the absence of time-reversal or space-inversion symmetry[1]. Of particular importance is that the integration of the Berry curvature over a surface enclosing one Weyl point yields the Chern number, giving rise to salient responses such as large anomalous Hall effect[2,3], negative magnetoresistance[4,5], and ultra-high conductance due to the suppression of backward-scattering processes[6]. Motivated by these remarkable properties,

recent extensive studies have revealed a number of topological quantum materials. Nonetheless, the control of the electronic states is still challenging, since the chemical substitution to control the band filling and/or band width may cause severe disorder effects in crystals.

Half-Heusler alloys $RTX$ ($R$ being a rare-earth, $T$ a transition metal, and $X$ a main-group element, shown in the inset of Fig. 1a) have attracted much attention since they possess a tunable electronic structure similar to zinc-blende (Hg,Cd)Te and further host magnetic

[1]Department of Applied Physics and Quantum Phase Electronics Center (QPEC), University of Tokyo, Tokyo 113-8656, Japan. [2]RIKEN Center for Emergent Matter Science (CEMS), Wako 351-0198, Japan. [3]Institute of Solid State Physics, University of Tokyo, Kashiwa 277-8561, Japan. [4]Department of Advanced Material Science, University of Tokyo, Kashiwa 277-8561, Japan. [5]Research Center for Advanced Science and Technology, University of Tokyo, Komaba Meguro-ku, Tokyo 153-8904, Japan. [6]Tokyo College, University of Tokyo, Tokyo 113-8656, Japan. ✉e-mail: ueda@ap.t.u-tokyo.ac.jp

**Fig. 1 | Basic properties of HoAuSn. a** The right-top inset in the center main panel shows the crystal structure of HoAuSn, composed of three face-centered cubic lattices of rare-earth $R$ (blue), transition metal $T$ (gold), and main-group element $X$ (silver), respectively. The center panel shows the schematic picture of the energy difference between the $\Gamma_6$ and the $\Gamma_8$ states as a function of $(Z_T + Z_X)V$, where $Z_T$ and $Z_X$ are atomic numbers of $T$ and $X$ atoms and $V$ is the unit-cell volume, drawn on the basis of ref. 9. The band order is trivial in materials located in the upper half (where $E_{\Gamma_6} > E_{\Gamma_8}$) whereas the band inversion occurs in materials located in the lower half. The top (bottom) panel exhibits the schematic band structures in materials located in the upper (lower) half. The exchange field lifts the band degeneracy and induces Weyl points. **b** Resistivity as a function of temperature. **c** Magnetization for the magnetic field along [111] as a function of temperature. **d** Specific heat of HoAuSn and LuAuSn as a function of temperature. **e** Integrated intensity of the magnetic (0.5,0.5,0.5) and nuclear (1,1,1) neutron diffraction peaks as a function of temperature. Error bars indicate the standard error derived from the square roots of the numbers of counts. The inset shows the scans along $(h, h, h)$ at 1.65 K. **f** Spin-polarized neutron scattering of the (0.5,0.5,1.5) peak at 1.65 K. Red and blue symbols denote the non-spin-flip (NSF) scattering and spin-flip (SF) scattering data, respectively. The red curve is a gaussian fit to the NSF data. The left inset shows the scattering geometry where $\vec{q}$ denotes the magnetic propagation vector and $\vec{Q}$ denotes the neutron scattering vector, respectively. $p_N$ is the direction of neutron spin polarization, which is perpendicular to the scattering plane ($h\,h\,l$). Open and filled squares show the reciprocal lattice points where the nuclear reflections are forbidden and allowed, respectively. The directions of Ho moments in the anti-ferromagnetic phase are shown in the right inset.

$R$ ions which can amplify the external magnetic-field effect on the conduction electrons[7,8]. Based on the band alignment ($\Gamma_8$ and $\Gamma_6$) at $k = 0$, the classification of $RTX$ compounds is suggested by former studies[7–9], as shown in Fig. 1a. Among them, GdPtBi, which is located in non-trivial region depicted in Fig. 1a (lower half), is a band-inverted zero-gap semiconductor in its paramagnetic state while magnetism arises from the Gd $4f$ electrons. External magnetic fields align the $4f$ moments and thereby cause the strong exchange band splitting of the $\Gamma_8$ band, resulting in the emergence of linearly crossing Weyl points, as depicted in the bottom panel of Fig. 1a[10–12]. Here, a natural question arises; does the exchange field give rise to Weyl points also in the non-band-inverted semiconducting state, as shown in the top panel of Fig. 1a? However, experiments have so far been limited to a few groups of materials such as topologically nontrivial $R$PtBi and $R$PdBi[10–13]. To expand the series of $RTX$ and understand the physics of band topology more deeply, the investigation of the unexplored electronic phases or materials, for instance, $R$AuSn located in the upper half of Fig. 1a, is highly desired.

In this study, we synthesize single crystals of the normal semimetal HoAuSn which has never been reported. We find that the external magnetic field induces a significant decrease in resistivity of more than four orders of magnitude, accompanied by the change of the electronic state. The field-induced semimetallic state exhibits the markedly large ratio (~1000) of the room-temperature to the residual resistivity and the small residual resistivity of ~0.2 $\mu\Omega$ cm, which are distinctive from those in iso-structural compounds. Our analysis combined with quantum oscillations and first-principles calculations reveals that the exchange band splitting gives rise to Weyl points, resulting in the strong suppression of the carrier backscattering. These findings suggest that unprecedented multifunctional topological phenomena can be hidden in unexplored $RTX$ materials.

## Results

### Crystal growth and characterization

According to a previous study on the structure of $RTX$ compounds[14], the cubic lattice structure (Fig. 1a) is stable for heavy $R$ compounds while the hexagonal one is realized in the case of light $R$ elements (Fig. S2a). HoAuSn is located on the boundary between these phases. We find that, in the present case of Sn flux method, the formed crystal structure of HoAuSn depends on the thermal condition; the cubic structure is stable at low temperatures whereas the hexagonal one is stabilized at high temperatures (see Supplementary Note I for details). Here we focus on a sample with the former crystalline structure (for the detailed characterization including powder x-ray diffraction and Laue pattern, see Supplementary Fig. S1). Figure 1b, c shows the temperature dependence of resistivity $\rho_{xx}$ and magnetization $M$. The resistivity exhibits the semiconductor like temperature dependence at high temperatures. With decreasing temperature below 20 K, the resistivity turns to slight decrease and there is discerned an anomaly at around $T_N = 1.9$ K, at which the magnetization exhibits a kink, similar to other antiferromagnetic half-Heusler compounds[15]. Figure 1d displays the temperature dependence of the specific heat for HoAuSn as well as non-magnetic LuAuSn as a reference for the lattice contribution. HoAuSn features an explicit peak at $T_N$, indicative of the second-order magnetic transition. A hump-like structure is centered at around 20 K, which is attributed to the Schottky anomaly due to the Ho $4f$ $J$ crystal-electronic-field splitting, as also observed in isostructural HoPdBi[16]. Incidentally, the first excited state is located at around 20 K, below which the resistivity calmly decreases as seen in Fig. 1b, presumably implying that the Ho $4f$ magnetism is correlated to the charge transport.

To verify the magnetic structure of these newly synthesized HoAuSn single crystals, we performed unpolarized and polarized

neutron scattering measurements. We first measured unpolarized neutron scattering profiles and found magnetic Bragg peaks which can be indexed by using magnetic propagation vectors ($q$-vectors) of (0.5,0.5,0.5) and its equivalents below $T_N$. No other peak except for nuclear Bragg peaks is observed on the high-symmetry axes (see Supplementary Fig. S4). Figure 1e shows the temperature dependence of the integrated intensity for the nuclear and the magnetic Bragg peaks at (1,1,1) and (0.5,0.5,0.5), respectively. The intensity of the magnetic peak gradually increases as the temperature decreases below $T_N$ whereas the intensity of (1,1,1) remains nearly intact. To determine the directions of the magnetic moments below $T_N$, we performed polarized neutron scattering measurements (see details in Supplementary Note 2). The direction of the neutron spin polarization was set to be perpendicular to the horizontal scattering plane, which was the $(h,h,l)$ plane. Figure 1f shows the rocking curve at (0.5,0.5,1.5), at which the scattering vector $\mathbf{Q}$ is nearly perpendicular to the q-vector of $\mathbf{q} = (0.5,0.5,-0.5)$ as shown in the inset. This geometry allows us to detect the Fourier-transformed spin components parallel and perpendicular to $\mathbf{q}$ in the spin-ip (SF) and non-spin-ip (NSF) scattering channels, respectively. We found that the SF scattering is almost negligible whereas the NSF scattering shows a clear gaussian-like peak, indicating that the spin orientation is perpendicular to $\mathbf{q}$. Thus, the Ho moments point ferromagnetically in the (111) planes which are in turn stacked antiferromagnetically along $\mathbf{q}$ (c.f. the right panel of Fig. 1a), as also seen in some other heavy rare-earth isostructural materials[17].

## Colossal negative magnetoresistance in longitudinal geometry

We now turn to the magnetotransport properties for HoAuSn. The sample geometry is shown in the inset of Fig. 2. The electric current I and the magnetic field $H$ are parallel to the [1$\bar{1}$0] crystalline direction, enabling us to measure the longitudinal magnetoresistance (MR). The sample plane parallel to the current is (111). Figure 2 shows the temperature dependence of the resistivity at several magnetic fields (Note that the ordinate is on a logarithmic scale). Below 50 K, the resistivity at $\mu_0 H = 1$ T is somewhat smaller than at 0 T, and sharply drops below 10 K. This peak-like structure shifts towards higher temperature while the resistivity systematically decreases as the field increases. Eventually, at 14 T, the resistivity significantly decreases below 100 K by three orders of magnitude ($\rho_{xx}$(100 K)/$\rho_{xx}$(2 K) = 1463), reaching quite a low residual resistivity, $\rho_0 = 0.18$ $\mu\Omega$ cm at 2 K. Such an excellent metallicity is distinct from other isostructural (half-)Heusler compounds[18] and rather comparable to topological semimetals with high-mobility carriers such as NbP[19], WTe2[20, 21], and Cd3As2[22]. The inset of Fig. 2 shows the magnetization dependence of the MR ratio $\rho_{xx}(M) = \rho_{xx}(0)$ at several temperatures, which gradually decreases with lowering temperature. Below 20 K, $\rho_{xx}(M)/\rho_{xx}(0)$ markedly increases at small $M$, and abruptly drops at around 1 $\mu_B$/f.u. The positive MR at low fields may be attributed to the Zeeman splitting of the Ho ground-state multiplet, which revives the fluctuation of $4f$ electrons and hence causes the carrier scattering.

There are several possible scenarios to explain the observed colossal negative magnetoresistance. For instance, the relation $\rho_{xx}(M)/\rho_{xx}(0) = 1$-$C(M/Ms)^2$ (where $C$ is a constant and $Ms$ is the saturated value of $M$) is suggested by the s-d model or Kondo lattice model to explain the transport properties in double-exchange ferromagnets[23, 24], and confirmed to explain well the observed negative magnetoresistance in the perovskite manganites as $C \sim 2$–$4$[25]. This model presumes that the negative MR solely depends on the strong (Hund's rule) coupling between the conduction electrons and the local moments. As shown in the inset of Fig. 2, the curve above 100 K can be described by this relation ($C \le 4$), but it deviates quickly as the temperature is lowered, pointing to the increasing importance of qualitative field-induced changes in the electronic structure in the present compound. Another examples of negative magnetoresistance effects are those observed in Eu oxides and chalcogenides[26–28], or dilute magnetic

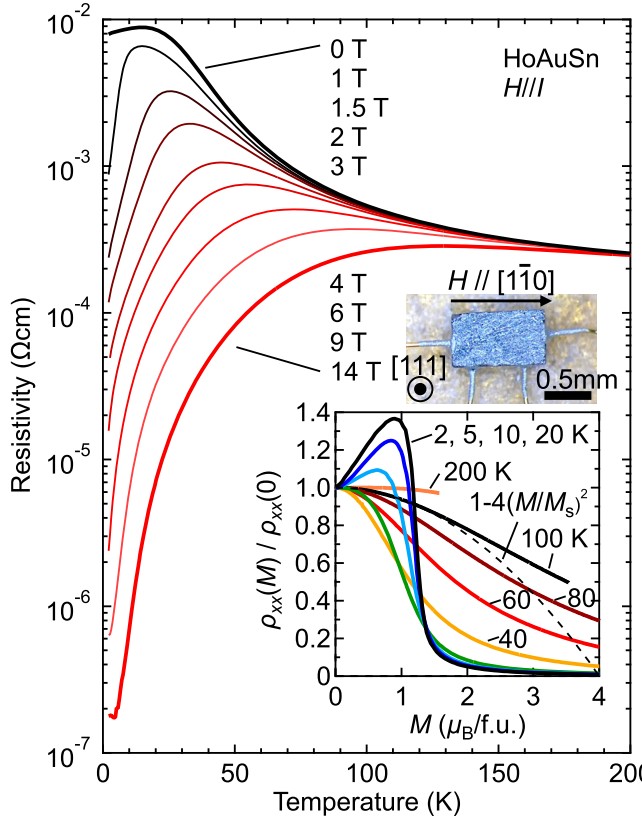

**Fig. 2 | Longitudinal magnetoresistance of HoAuSn.** The temperature dependence of the longitudinal resistivity $\rho_{xx}$ at several magnetic fields. The upper inset shows the sample geometry where the electric current and the magnetic field are along [1$\bar{1}$0]. The lower inset shows the magnetization ($M$) dependence of the resistivity ratio $\rho_{xx}(M) / \rho_{xx}(0)$ at several temperatures. The dashed line denotes the relation that $\rho_{xx}(M) / \rho_{xx}(0) = 1$- $C(M \cdot Ms)^2$, with $C = 4$ and $Ms$ being the saturation magnetization.

semiconductors[29,30]. These effects are interpreted in terms of several mechanisms such as spin-disorder scattering and the formation of bound magnetic polarons. However, it is worth noting that the resistivity of the ferromagnetic metal phase in these materials is quite large (>1 mΩ cm) compared to that in the present case (<1 μΩ cm). This suggests that the ultrahigh mobility is realized at high fields and thus the origin of the field-induced negative magnetoresistance in HoAuSn is quite unique. A more plausible origin is the exchange splitting of the band via the intra-atomic magnetic coupling between Ho $5d$ and $4f$ electrons. According to the fat band calculation, the $R$ $5d$ orbitals are strongly hybridized with the conduction bands especially near the Fermi level[31]. Therefore, as the external field aligns the local Ho $4f$ moments, the exchange field can lift the spin degeneracy of the electronic bands and induce significant changes of the Fermi surfaces. Similar mechanism of the electronic-band reconstruction is proposed in the isostructural GdPtBi which becomes Weyl semimetal from the zero-gap semiconductor by small applied fields[11,12].

## Transverse magnetotransport properties

To elucidate this scenario, we measured transverse magnetotransport properties ($H$//[111], $I$//[1$\bar{1}$0]). Figure 3a shows the magnetic field dependence of the resistivity at several temperatures. The field dependence of the transverse MR becomes apparent below 80 K. At 0.5 K, the resistivity sharply decreases at around 1 T, turns to increase at around 3 T, and gradually increases towards $\sim 1.2 \times 10^{-3}$ Ω cm at 14 T, which is more than four orders of magnitude larger than the longitudinal MR for $H$//$I$. This remarkable difference between the

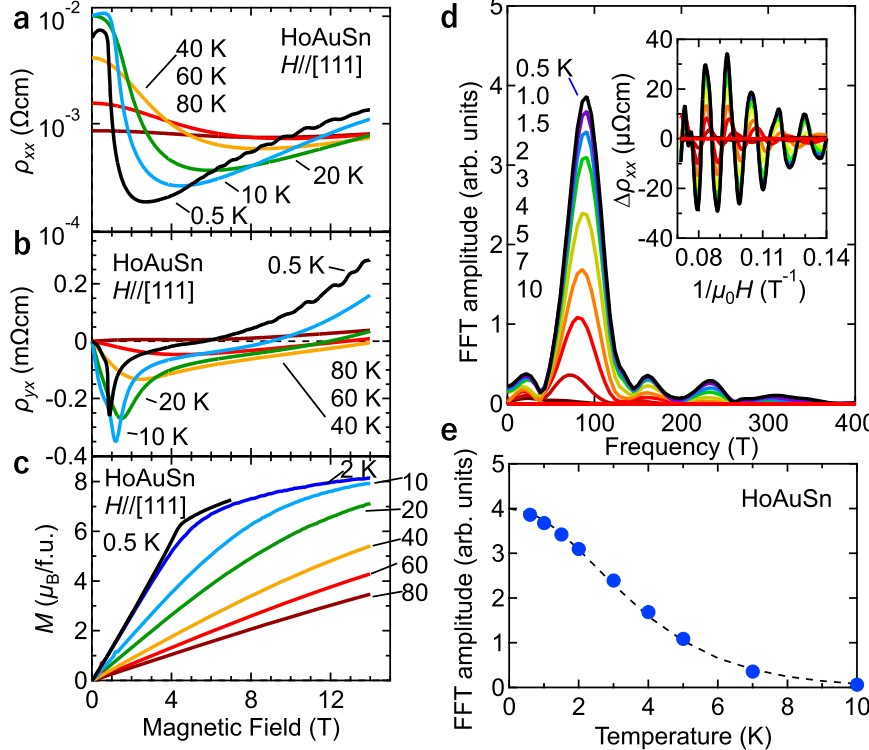

**Fig. 3 | Magnetotransport properties and quantum oscillations in HoAuSn.**
**a** Transverse ($H \perp I$) magnetoresistivity with the current ($I$) path along [1$\bar{1}$0] as a function of magnetic field ($H$) along [111] at several temperatures. **b** Hall resistivity as a function of magnetic field along [111] at several temperatures. **c** Magnetization curves with $H$ along [111] at several temperatures. **d** Fast-Fourier transformation amplitudes as a function of frequency. In the inset, the oscillatory part of the resistivity after subtracting the background (see text) is plotted as a function of $1 = \mu_0 H$ for each temperature. The peak amplitudes are damped with increasing temperature (see panel d for the color code). **e** Amplitude variation of the peak as a function of temperature. The dashed line is a fit to the data.

longitudinal and transverse MRs can be attributed to the orbital motion of electrons, as typically observed in high-mobility compensated semimetals[32]. In short, in the transverse MR configuration, the negative MR due to the band-structural change from the exchange field and the positive MR due to the field-induced orbital motion competes and partly cancels each other, resulting in the non-monotonic MR as observed. Figure 3b shows the Hall resistivity $\rho_{yx}$ at several temperatures; $\rho_{yx}$ is small and positive above 80 K. Upon decreasing temperature, the sign reverses and a broad dip structure shows up. Remarkably, $\rho_{yx}$ at 0.5 K exhibits a sharp drop at around 1 T at which the resistivity for both $H//I$ and $H \perp I$ abruptly decreases as well. With further increasing field, $\rho_{yx}$ largely increases and reaches the positive value of 0.3 mΩcm at 14 T. We plot the temperature dependence of Hall resistivity at 1 T and 14 T in Supplementary Fig. S3. The Hall resistivity at 1 T significantly decreases with lowering temperature below 80 K, whereas those at 14 T sharply increases. In contrast, the magnetization exhibits a monotonic field and temperature dependence (Fig. 3c). At 0.5 K, the magnetization features a small kink at around 4 T and gradually increases up to almost 8 $\mu_B$/f.u. Therefore, the observed non-monotonic behavior in $\rho_{yx}$ at low temperatures can be due to the drastic change of the Fermi surfaces (i.e. Lifshitz transition), rather than the magnetic transitions. It is in fact consistent with our calculation which demonstrates the evolution of the band structure as the Ho magnetic moments gradually aligns along the field direction (see Supplementary Note V and Fig. S11 for details). It is noteworthy that the similar behaviors of magnetotransport shown in Fig.2 (longitudinal configuration, $H//I$) and 3 (transverse configuration, $H \perp I$) are discerned respectively in the same, longitudinal or transverse, configuration but irrespective of the crystalline geometry (see the Supplementary Note IV and Figs. S7 and S8 for details).

Next we investigate the Fermi surface topology by analyzing the Shubnikov-de Haas (SdH) oscillatory components $\Delta\rho_{xx}$ in resistivity (Fig. 3a) by fitting the non-oscillatory part with a polynomial function of the field and subtracting it from $\rho_{xx}$. The inset of Fig. 3d shows $\Delta\rho_{xx}$ as a function of the inverse magnetic field. Clear beating patterns with systematic temperature dependences are observed below 10 K. Figure 3d summarizes the fast-Fourier transform (FFT) of $\Delta\rho_{xx}$ at each temperature, revealing a clear peak at 90.5 T at 0.5 K as well as several small peaks below 300 T. The multifrequencies with small intensities may be associated with the lifting of degenerate bands as observed in the calculated band structures discussed below. We also derive the cyclotron masses of $m^* = 0.40m_0$ by fitting the temperature dependence of the oscillation amplitude with the Lifshitz-Kosevich formula, as shown in Fig. 3e.

**First-principles band calculations**

We performed first-principles band calculations for HoAuSn within the framework of the density functional theory (DFT), for several magnetic ground states of the Ho 4f moments in the presence of spin-orbit coupling; a non-magnetic (NM), an antiferromagnetic (AFM), and a ferromagnetic (FM) ground state, respectively, as shown in Fig. 4. The bottom of the conduction band is located at the zone boundary while the top of the valence band is near the Γ point for all magnetic states, consistent with the previous study on nonmagnetic LuAuSn[31]. While the band structure of the NM state seems to be similar to that of AFM, the band dispersion of FM is distinct from the other two. The spin polarization in FM causes a large band splitting and, hence, expands the Fermi surface cross section. Furthermore, there are several degenerate points between the second and third highest valence bands along the Γ-L direction as magni ed in the inset of Fig. 4c. The calculated value of kF or the cross section for the hole pocket $\alpha_2$

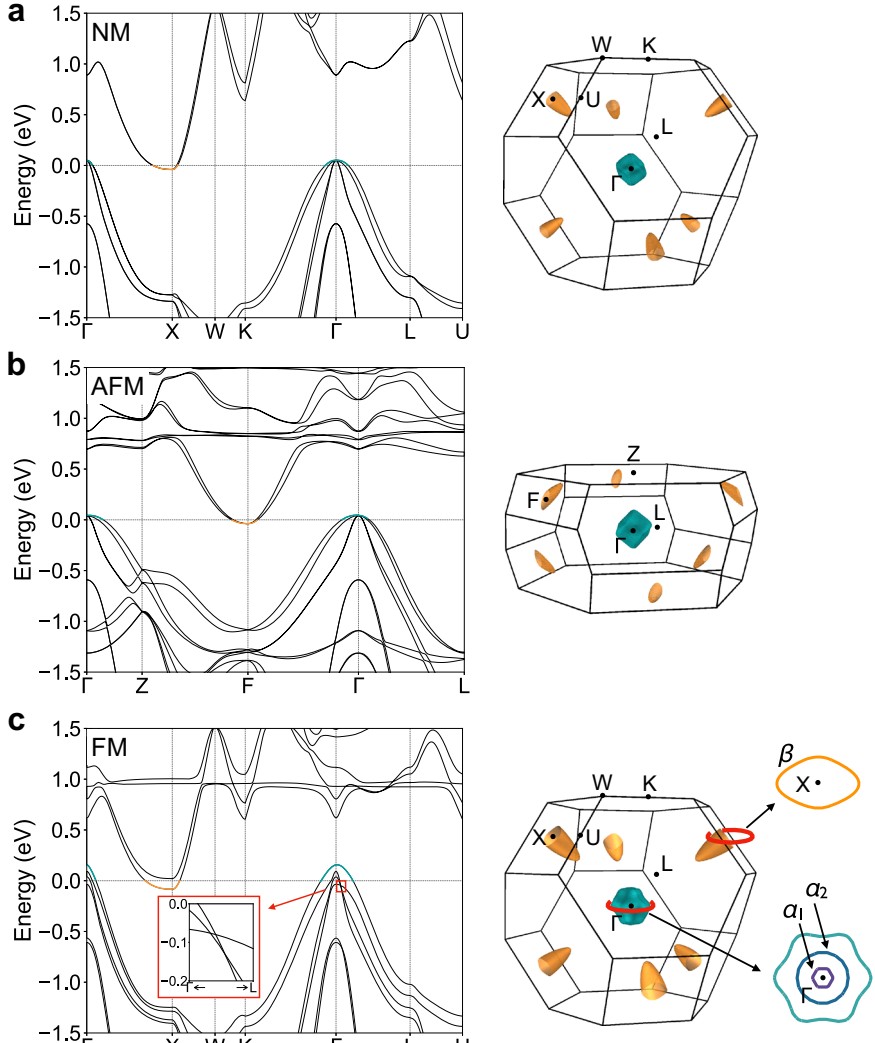

**Fig. 4 | Electronic band structures of HoAuSn with several magnetic ground states.** Energy dispersed band structure with spin-orbit coupling along the high symmetry axes (left panels) and the shape of the Fermi surfaces of the highest valence band and the lowest conduction band in the first Brillouin zone (right panels). Yellow bullet-like electron pockets are located at the zone boundary while green hole pockets are at the zone center. **a** Non-magnetic (NM) state.

**b** Antiferromagnetic (AFM) state. **c** Ferromagnetic (FM) state. The right schematics exhibit the Fermi surface cross sections of the hole and electron pockets in the (111) plane. The smallest, second smallest hole pockets, and electron pocket correspond to $\alpha_1$, $\alpha_2$, and $\beta$ which are revealed by the quantum oscillation in Fig. 3d, respectively.

around Γ in FM is in reasonably good agreement with that estimated from the SdH oscillations. Note that the other small Fermi surfaces such as the hole pocket $\alpha_1$ and the electron pocket $\beta$ may be observed as the peaks with small intensity in Fig. 3d, and the outermost hole pocket could not be observed in SdH due perhaps to the limited field range of the present study (see Supplementary Note V for details).

## Discussion

Having established the band structure in each magnetic state, we discuss the origin of the negative MR shown in Figs. 2 and 3. According to our calculation, the Fermi surface cross section in the FM state is about twice as large as that in NM or AFM. However, as shown in Fig. 2, the resistivity decreases by more than four orders of magnitude as a function of the magnetic field, which cannot be explained solely by the change of the carrier concentration on the basis of the simple Drude model ($\sigma_{xx} = ne^2\tau/m^*$ where n is the carrier concentration and $\tau$ is the scattering relaxation time). Therefore, $\tau$ should be altered dramatically in the FM state. In fact, the quantum oscillation is observed as in Fig. 3d, indicating that of the hole carriers is markedly pronounced. To

clarify the origin of the enhancement of $\tau$, we identify the chirality of the degenerate points between the second and third highest valence bands along the Γ-L direction, which turn out to be Weyl points (see details in Supplementary note V and Figs. S9 and S10). The Berry phase induced by Weyl points can effectively suppress backscattering, thereby resulting in a notable enhancement of the relaxation time $\tau$. In addition, the linear-like dispersions around Weyl points contribute to the generation of high mobility and a substantial reduction in residual resistivity. We provide more detailed analysis and supporting calculations in Supplementary Note V. It is known that the Berry phase induced by Weyl points can effectively suppress the backscattering, leading to the ultra-high mobility and considerably low residual resistivity[6, 22]. In fact, the mean $\tau$ averaged over all carriers is estimated as $8.0 \times 10^{-11}$ s (see Supplementary note V for details), comparable to Dirac/Weyl semimetals such as WTe$_2$ ($5.5 \times 10^{-11}$ s[21]), Cd$_3$As$_2$ ($2.1 \times 10^{-10}$ s[22]), or TaAs ($2.6 \times 10^{-11}$ s[33]). Future spectroscopic studies which can directly observe the electronic states will serve to test the mechanism of the observed mobility enhancement. Our observation reveals that magnetic half-Heusler compounds are an ideal group of

materials to study field-controllable Weyl semimetals, and demonstrates a unique mechanism of magnetoresistance which is also important for connecting topological physics and spintronics applications.

## Methods

### Single crystal growth

Single crystals of HoAuSn were grown using a Sn self-flux method. A mixture of Ho pieces (with a purity of 99.9%), Au powder (99.9%), and Sn grains (99.999%) in the molar ratio of 1:1:15 was put into an alumina crucible and sealed inside an evacuated quartz tube. The ampoule was heated to 1000 °C and kept for 24 h to obtain a homogeneous solution. The samples were cooled at a rate of 2 °C/h and centrifuged at 500 °C to dissociate the HoAuSn crystals from the Sn flux. We obtained two types of single crystals as described in Supplementary note 1. We characterized both types by x-ray diffraction, transport, and magnetization measurements. Finally we obtained the single crystals of half-Heusler type HoAuSn with no trace of impurity.

### Transport, specific-heat, and magnetization measurements

Transport and specific heat were measured by a conventional four-probe method and a relaxation method, respectively, in a Physical Property Measurement System equipped with a rotator and $^3$He insert (Quantum Design). In the transport measurements Au wires were attached to the sample with Ag paste (DuPont 4922 N). The typical dimensions were 1.2 mm × 0.5 mm × 0.1 mm (shown in the inset of Fig. 2), which enable us to thoroughly measure the transport properties for both longitudinal and transverse configurations. The magnetic field dependence of resistivity (Hall resistivity) is obtained after the symmetrization (antisymmetrization) with respect to the positive and negative field. Magnetization was measured with a Magnetic Property Measurement System MPMS3, equipped with a $^3$He insert (Quantum Design).

### Quantum oscillation

The Shubnikov-de Haas oscillatory component $\Delta\rho_{xx}$ in resistivity was deduced by fitting the non-oscillatory part with a polynomial function of the magnetic field. The obtained frequency F is proportional to the corresponding extremal Fermi surface cross-section $A_{\text{ext}}$ that is perpendicular to the magnetic field, satisfying the Onsager relation $F = (\Psi_0/2\pi^2)A_{\text{ext}}$ with $A_{\text{ext}} = \pi k_F^2$, where $\Psi_0 = h/2e$ is the magnetic flux quantum, $h$ is the Planck constant, and $k_F$ is the Fermi wave vector. The cyclotron mass was deduced by fitting the temperature dependence of the oscillation amplitude with the Lifshitz-Kosevich formula.

### Neutron scattering measurements

Neutron scattering experiments were carried out on the polarized neutron triple-axis spectrometer (PONTA) at 5 G beamline of Japan Research Reactor 3 (JRR-3) in Japan. An unpolarized monochromatic neutron beam with the energy of 14.7 meV was obtained using a pyrolytic graphite (PG) monochromator. The single crystal sample was loaded into a pumped $^4$He cryostat with the (h,h,l) scattering plane. The data shown in Fig. 1e were measured in the unpolarized condition without analyzer. For polarized measurements, we employed a supermirror spin polarizer and Heusler (111) crystal analyzer. The direction of neutron spin polarization at the sample position was set to be perpendicular to the scattering plane, namely [1$\bar{1}$0] direction, by a Helmholtz coil. We measured SF and NSF scatterings by changing the spin state using a spin flipper placed between the sample and the polarizer. The flipping ratio of the neutron beam was 14.42. For Fig. 1f, the effect of the imperfect beam polarization was corrected.

### First-principles calculations

The DFT calculations are performed on the basis of the projector augmented wave scheme[34] as implemented in the Vienna ab initio simulation package (VASP)[35]. We employ the exchange-correlation functional given by the generalized gradient approximation (GGA) with the Perdew-Burke-Ernzerhof (PBE) type parametrization[36]. Spin-orbit coupling is switched on. Strong intra-atomic interactions in 4f orbitals of Ho are approximated by the GGA + U method[37], with U = 6.7 eV and J = 0.7 eV[38]. To avoid possible underestimation of the band gap, we also perform the calculation with the HSE06 functional[39], in addition to the GGA + U formalism. In particular for the important FM state, HSE06 confirms the negative indirect gap as predicted by GGA + U (−0.124 eV and −0.241 eV, respectively), qualitatively in agreement with the semimetallic nature observed in experiment. The energy cutoff of the plane wave is chosen as 370 eV. The Brillouin zone is sampled by a Γ-centered 8 × 8 × 8 grid. The Weyl points are searched on the basis of the Wannier interpolated band structure[40–42]. The chirality of each Weyl point is computed by integrating the Berry flux over a surface enclosing the Weyl point[42–44]. We use pre-/post-processing tools and utilities[45–49] to aid the calculations.

## Data availability

The data that support these findings are available from the corresponding authors upon request.

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

## Acknowledgements

This work was supported by JSPS Grant-in-Aid for Scientific Research (No. 21K13871), and CREST (No. JPMJCR16F1 and JPMJCR1874), Japan Science and Technology Japan. The neutron scattering experiment in JRR-3 was performed under user programs (Proposal No. 22517).

## Author contributions

Y.T. and K.U. conceived the project. K.U. and R.K. performed the crystal growth. K.U., M. Hoshino, and D.H. performed the x-ray diffraction and analysis. K.U. carried out resistivity, magnetization, and specific-heat measurements. K.U. and M.K. extended the temperature range to 3He temperatures. K.U., R.K., T.N., H.S. and T.A. performed the neutron diffraction experiments. T.Y., M. Hirayama, and R.A. performed the first-principles band calculation. K.U., T.Y. and Y.T. wrote the manuscript. All authors contributed to the scientific planning and discussions.

## Competing interests

The authors declare no competing interests.
