## [Peer Review File · Nature Communications]

REVIEWER COMMENTS

Reviewer #1 (Remarks to the Author):

Unfortunately, the manuscript is not ready for publication, and the data do not support the claims. My main objections follow.

The Authors do not specify the samples' geometry to determine the resistivity tensor components. The drawings suggest that the samples had a square shape. If this was the case, the geometrical magnetoresistance would preclude a meaningful determination of ρ_{xx} and ρ_{yx} in the magnetic field $\mu B > 1$, the case of the samples under consideration, as the Shubnikov-de Haas oscillations were observed. In other words, relatively large values of ρ_{xx} compared to ρ_{zz} and a sign change of ρ_{yx} in high magnetic fields originate instead from the sample shape than from the physics of the material.

I am afraid that given the number of Shubnikov de Haas oscillations observed, the interpretation of the data in terms of three cross-sections appears as over interpretation.

Since its discovery in Eu oxides and chalcogenides in the 1960s, giant negative magnetoresistance has been observed in numerous magnetic and dilute magnetic semiconductors. One of the examples is, for instance, presented by Liu et al, PRL 108, 036805 (2012) – see also references therein.

It is not clear whether first-principles calculations take into account spin-orbit splitting. If not, comparing experimental results to computed cross-sections of the Fermi spheres is misleading. Furthermore, in the employed computation scheme, the DFT underestimates energy gaps. Therefore the semimetallic band arrangement predicted theoretically can be a computational artifact.

Reviewer #2 (Remarks to the Author):

The Authors did not adhere to the format of the Journal. For example, after the Abstract, no other section was highlighted. May I suggest following the laid down format?

The title is appropriate and reflects the current topic in the field. Heusler Alloys are very important classes of compounds in modern technological applications. Therefore, the manuscript will arouse the interest of many researchers in Materials Science, Physics, and interdisciplinary research.

The other parts of the manuscript are well-written in clear English Language.

I, therefore, recommend publication.

Reviewer #3 (Remarks to the Author):

The authors present a combined experimental and theoretical study of the proposed field-induced Weyl semimetal of magnetic half-Heusler compound HoAuSn. Overall, I find that the data is of a high quality and shows a number of interesting findings. The conclusions are at present however not sufficiently well justified or explained, and I have a number of questions that I would wish to see addressed before any possible publication of this work.

1) One of the key results in this work is to obtain the cubic half-Heusler single crystals HoAuSn. However, very little information is given about the quality of the single crystals. In particular, it is important to confirm the structure and also the orientation of crystals by using Laue X-ray diffraction pattern or other techniques. Actually, it is very interesting to us, how and why both cubic and hexagonal lattice structure HoAuSn single crystals can be obtained by using Sn flux method? Moreover, it is necessary to check the composition of the studied single crystals in this work. Are these magnetic and magnetotransport properties were measured using the same crystals? How many single crystals have been measured in this work?

2) It is unclear from the paper how the authors believe that the electronic structure is evolving with applied field, e.g., field-induced reconstructs the band structure and how this gives rise to the observed colossal negative magnetoresistance? In Fig. 2, the author only show the temperature dependence of the longitudinal resistivity with the electric current and the magnetic field are along [1-10] direction, and how about other geometries? Is the the colossal negative magnetoresistance relies on the direction of crystals? or more substantial deviations from the sample geometry measured in that study? Given the importance of the transverse ($H \perp I$) magnetoresistivity shown in Fig. 3, different measuring geometry must be performed.

3) The paper would strongly benefit from a much more thorough discussion of how the colossal negative magnetoresistance are expected to arise from the field-induced reconstructs the band structure, and how the authors believe that this evolves with applied field, supported by more

detailed analysis of the Shubnikov-de Haas oscillations and first-principles calculation. However, it has been theoretically predicted some time ago and recently been confirmed that magnetic half-Heusler compounds may show an unusual electronic structure with field-induced Weyl point and that the violation of time reversal symmetry by ferromagnetism and spin-orbit coupling will lead to a more complicated band structure (see Refs. 4 and 5). In this context the interpretation of the electronic band structures of HoAuSn (Fig. 4 and accompanying text) is rather vague.

4) The identification of the several magnetic ground states from first-principles band calculations is convincing. From this, they argue that the colossal negative magnetoresistance are expected to arise from the field-induced reconstructed band structure. The Berry phase induced by Weyl points can effectively suppress the backscattering, leading to the ultra-high mobility. This is perhaps plausible, but is not sufficiently justified. In other words, the study would be further improved by some theoretical support for the proposed interpretation at least the distribution of the Weyl points at FM states shown in Fig. S6 and also their evolution with the external magnetic fields.

5) Furthermore, I didn't find any information about the Hall data regarding the electron and hole carriers predicted from the first-principles band calculations. It is very important to exclude the contribution of both the simple Drude model and the multi-band model, if the author argue the the origin of the negative MR is due to field-induced Weyl point at HoAuSn.

Overall there are interesting features in the experimental data here. An appealing assignment of them is presented but to my mind a substantially enhanced justification and explanation of these would be required prior to publication.

Authors' response to the Reviewer #1

Unfortunately, the manuscript is not ready for publication, and the data do not support the claims. My main objections follow.

First of all, we thank the Referee for his/her kind review of the manuscript. Following the Reviewer's comment, we have revised the manuscript and Supplementary Information, which we believe are much improved. Our reply to each comment is as follows.

(1-1) comment

The Authors do not specify the samples' geometry to determine the resistivity tensor components. The drawings suggest that the samples had a square shape. If this was the case, the geometrical magnetoresistance would preclude a meaningful determination of ρ_{xx} and ρ_{yx} in the magnetic field $\mu B > 1$, the case of the samples under consideration, as the Shubnikov-de Haas oscillations were observed. In other words, relatively large values of ρ_{xx} compared to ρ_{zz} and a sign change of ρ_{yx} in high magnetic fields originate instead from the sample shape than from the physics of the material.

(1-1) reply

To avoid misleading the sample shape, we have replaced the drawing in Fig. 2 with the picture of the actual sample and added the information of the sample dimensions in Methods. We have also added the sample dependence of the transport properties in Fig. S3 in Supplementary Information. All of them show the large negative magnetoresistance at low temperatures. Furthermore, we have added the field angle dependence of resistivity for two samples with different crystalline geometry in Fig. S8, which shows the systematic change as a function of the angle. These results ensure that the observed transport properties are intrinsic.

We have revised the Fig. 2 and added the description of the typical sample dimensions in Methods. We have also added the last subsection in Supplementary Note I, the sentences in Supplementary Note IV, and Supplementary Figs. S3 and S8.

(1-2) comment

I am afraid that given the number of Shubnikov de Haas oscillations observed, the interpretation of the data in terms of three cross-sections appears as over interpretation.

(1-2) reply

We have deleted the assignment of α_1 and β in Fig. 3d and 3e in the main text. Instead, we have used the data in Supplementary Note V where we estimate the relaxation time of the field-induced semimetal state. Incidentally, the experimentally estimated Fermi wave vectors α_1 and β are consistent with the calculated values, even though some of the peak intensities in SdH oscillation are small. Hence we think that it is helpful to use the data for the rough estimation of the relaxation time.

We have revised the sentences at P.9 L165-169 and Figs. 3d and 3e in the main text, and added the explanations in Supplementary Note VI and Fig. S12.

(1-3) comment

Since its discovery in Eu oxides and chalcogenides in the 1960s, giant negative magnetoresistance has been observed in numerous magnetic and dilute magnetic semiconductors. One of the examples is, for instance, presented by Liu et al, PRL 108, 036805 (2012) – see also references therein.

(1-3) reply

We thank the Reviewer for the important comments on the magnetoresistance effect in magnetic semiconductors including Eu oxides and chalcogenides. We have added sentences on this effect at P. 7 L.115-122 and references [24-28], so that we can deepen our discussion of the mechanism related to the observed magnetoresistance.

The important point in the present result is that the resistivity in the field-induced semimetal state in HoAuSn is much smaller than those in the ferromagnetic phase of conventional magnetic semiconductors. Furthermore, HoAuSn shows the quantum oscillation above 6 T, indicative of high mobility that is distinctive from Eu oxides and chalcogenides. These indicate that the origin of the magnetoresistive effect in HoAuSn is completely different from the conventional one, as discussed in detail in the main text.

We have added the sentences at P.7 L.115-122 and references [24-28].

(1-4) comment

It is not clear whether first-principles calculations take into account spin-orbit splitting. If not, comparing experimental results to computed cross-sections of the Fermi spheres is misleading. Furthermore, in the employed computation scheme, the DFT underestimates energy gaps. Therefore the semimetallic band arrangement predicted theoretically can be a computational artifact.

(1-4) reply

We thank the Reviewer for pointing out a possible misunderstanding the reader might have. In all the first-principles calculations we took the spin-orbit coupling and the associated splitting explicitly into account. We have indicated the inclusion of spin-orbit coupling in the band calculation clearly in the main text, Methods, and the caption of Fig. 4.

As the Reviewer noted, DFT with the GGA functional could underestimate energy gap in general. In fact, we verified the semimetallic nature with the HSE06 hybrid functional, which is much more computationally expensive but better performed in evaluating band gaps. For the important FM state, the GGA+*U* method predicts a negative indirect gap -0.241 eV, which is qualitatively validated by the HSE06 calculation (with the gap -0.124 eV). Furthermore, the resistivity observed in experiment does not diverge but slightly decrease at low temperature, also supporting a closed band gap. These complementary explanations have been added to the band calculation part in Methods.

Authors' response to the Reviewer #2

The Authors did not adhere to the format of the Journal. For example, after the Abstract, no other section was highlighted. May I suggest following the laid down format?

The title is appropriate and reflects the current topic in the field. Huesler Alloys are very important classes of compounds in modern technological applications. Therefore, the manuscript will arouse the interest of many researchers in Materials Science, Physics, and interdisciplinary research.

The other parts of the manuscript are well-written in clear English Language.

I, therefore, recommend publication.

We thank the Referee for his/her kind review of the manuscript and approval for the importance of our findings. Following the suggestion, we have used the format, including sections.

Authors' response to the Reviewer #3

The authors present a combined experimental and theoretical study of the proposed field-induced Weyl semimetal of magnetic half-Heusler compound HoAuSn. Overall, I find that the data is of a high quality and shows a number of interesting findings. The conclusions are at present however not sufficiently well justified or explained, and I have a number of questions that I would wish to see addressed before any possible publication of this work.

...

Overall there are interesting features in the experimental data here. An appealing assignment of them is presented but to my mind a substantially enhanced justification and explanation of these would be required prior to publication.

We thank the Referee for constructive comments and approval for the importance of our findings. Following his/her advice, we have revised the text and added further explanations on crystal growth and characterization. Furthermore, we have added some band calculations so that we can discuss the origin of the observed magnetoresistance more clearly and convincingly. The following is our response to each comment.

(3-1) comment

One of the key results in this work is to obtain the cubic half-Heusler single crystals HoAuSn. However, very little information is given about the quality of the single crystals. In particular, it is important to confirm the structure and also the orientation of crystals by using Laue X-ray diffraction pattern or other techniques. Actually, it is very interesting to us, how and why both cubic and hexagonal lattice structure HoAuSn single crystals can be obtained by using Sn flux method? Moreover, it is necessary to check the composition of the studied single crystals in this work. Are these magnetic and magnetotransport properties were measured using the same crystals? How many single crystals have been measured in this work?

(3-1) reply

We have added the information on the crystal growth and characterization in Methods in the revised manuscript. In particular, we have added the growth condition more in detail in Supplementary Note I. We have found that the hexagonal HoAuSn is grown in the high temperature condition whereas the cubic one is stable at low temperatures as described in Methods. As for the characterization, firstly, we have added the Laue diffraction patterns in Fig. S1e and S1f. We note that the sample orientation is also confirmed by neutron diffraction, as shown in Fig. S4. Secondly, we have added the lattice constant which is reasonably close to those of isostructural HoPtBi and HoPdBi as reported in Refs. [3] and [4] in the revised Supplementary Information. Thirdly, we have added the sample dependence of the transport properties in Fig. S3, where we show five samples in the same geometry. All of them show the similar temperature and magnetic field dependence, except for the residual resistivity at 14 T, probably due to a slight variation of the sample quality.

We have revised the sentences at P.5 L.50-55 and Methods in the revised manuscript, added the last subsection in Supplementary Note I, and added the Refs. [3-4], Figs. S1e, S1f, and S3 in Supplementary Information.

(3-2) comment

It is unclear from the paper how the authors believe that the electronic structure is evolving with applied field, e.g., field-induced reconstructs the band structure and how this gives rise to the observed Colossal negative magnetoresistance? In Fig. 2, the author only show the temperature dependence of the longitudinal resistivity with the electric current and the magnetic field are along [1-10] direction, and how about other geometries? Is the the colossal negative magnetoresistance relays on the direction of crystals? or more substantial deviations from the sample geometry measured in that study? Given the importance of the transverse ($H \perp I$) magnetoresistivity shown in Fig. 3, different measuring geometry must be performed.

(3-2) reply

In isostructural half-Heusler GdPtBi, Weyl semimetal state is induced by the exchange splitting as the external field polarizes the rare-earth Gd magnetic moment. The magnitude of the band exchange splitting is estimated to be as large as 0.5 eV [5], which is large enough to be detected by the transport

measurements. Therefore, a comparably large band structural change can occur in our HoAuSn, in which the Ho magnetic moments are easily aligned by the external fields. We have added the explanation at P.7 L.128-130 in the revised manuscript.

As for the relation between the magnetotransport and crystalline geometry, similar magnetotransport properties are observed in different sample geometries as shown in Supplementary note IV and Fig. S7. In addition to these results, we have added the magnetic field angle dependence of two samples in Fig. S8. One is the sample which is used in the main text and the other is the sample shown in Fig. S6. The electric current direction is [1-10] and the magnetic field is rotated from [1-10] to [111] in the former sample (Fig. S8a) while the current direction is [100] and the magnetic field is rotated from [100] to [001] in the latter (Fig. S8b). The angle θ denotes the field direction against the current. Similar angle dependence is discerned in both samples. For instance, there is almost no angle dependence below 1.5 T. Above 1.5 T, the resistivity significantly decreases for $\theta = 0^\circ$ whereas the resistivity shows upturn for $\theta = 90^\circ$ due to the positive transverse-magnetoresistance (added to the colossal negative magnetoresistance) characteristic of the high-mobility carrier's cyclotron motion. The resistivity systematically changes with varying θ for both samples. These properties indicate that the magnetoresistance simply depends on the angle between the current and the field direction, i.e., longitudinal vs. transverse MR configuration, irrespective of crystalline geometry.

We have added the sentences at P.7 L.128-130 in the revised main text and added the explanations in Supplementary Note IV and Fig. S8 in the revised Supplementary Information.

(3-3) comment

The paper would strongly benefit from a much more thorough discussion of how the colossal negative magnetoresistance are expected to arise from the field-induced reconstructs the band structure, and how the authors believe that this evolves with applied field, supported by more detailed analysis of the Shubnikov-de Haas oscillations and first-principles calculation. However, it has been theoretically predicted some time ago and recently been confirmed that magnetic half-Heusler compounds may show an unusual electronic structure with field-induced Weyl point and that the violation of time reversal symmetry by

ferromagnetism and spin-orbit coupling will lead to a more complicated band structure (see Refs. 4 and 5). In this context the interpretation of the electronic band structures of HoAuSn (Fig. 4 and accompanying text) is rather vague.

(3-3) reply

We would like to express our gratitude to the Reviewer for the valuable suggestion. We entirely agree on the significant role of the field-induced reconstruction of the electronic structure in generating the observed colossal negative magnetoresistance and high electron mobility, and have placed greater emphasis on it in the improved manuscript. These improvements are reflected at P. 12 L.201-205 in the updated Discussion section of the main text, accompanied by more comprehensive analyses and references [7-8] presented in Supplementary Note V and Figs. S9, S10, and S11 in Supplementary Information. To enhance the clarity, we have included detailed discussions on the Weyl points and their associated Berry curvature distribution, providing a more concrete demonstration of the Berry phase and Weyl nature. Additionally, we have estimated and plotted the spin texture to explicitly illustrate the spin-momentum locking effect induced by the Weyl points in Figs. S9 and S10, which significantly suppresses electron scattering. Furthermore, we have conducted band structure calculations to illustrate the evolution of the band structure and Weyl points in response to the application of an external field as shown in Fig. S11. These additional analyses have granted us a deeper understanding of the intricate electronic structure as reconstructed by the external field.

(3-4) comment

The identification of the several magnetic ground states from first-principles band calculations is convincing. From this, they argue that the Colossal negative magnetoresistance are expected to arise from the field-induced reconstructs the band structure. The Berry phase induced by Weyl points can effectively suppress the backscattering, leading to the ultra-high mobility. This is perhaps plausible, but is not sufficiently justified. In other words, the study would be further improved by some theoretical support for the proposed interpretation at least the distribution of the Weyl points at FM states shown in Fig. S6 and also their evolution with the external magnetic fields.

(3-4) reply

We sincerely appreciate the helpful suggestion provided by the Reviewer. Like the comment (3-3), the Reviewer hoped to see that we provide more convincing argument on the contribution of field-reconstructed band structure to the colossal negative magnetoresistance. In response, we have made significant improvements to our manuscript, as summarized in the reply to (3-3). Specifically, to evidence the suppression of electron scattering from the Berry phase, we have presented the distribution and evolution of Weyl points as suggested by the Reviewer, and in addition the Berry curvature and spin texture, in Supplementary Note V.

(3-5) comment

Furthermore, I didn't find any information about the Hall data regarding the electron and hole carriers predicted from the first-principles band calculations. It is very important to exclude the contribution of both the simple Drude model and the multi-band model, if the author argue the the origin of the negative MR is due to field-induced Weyl point at HoAuSn.

(3-5) reply

As shown in Fig. 3b, the Hall resistivity at low temperatures shows unique magnetic field dependence. Additionally, we plot the temperature dependence of Hall resistivity at 1 T and 14 T in Fig. S6 in the Supplementary Information. In particular, the absolute value becomes significantly large with lowering temperature below 80 K, accompanied by the sign change between 1 T and 14 T. These behaviors can be hardly explained by the conventional Drude or multi-band model. It can be rather attributed to the drastic change of the Fermi surfaces induced by the exchange band splitting. In fact, as shown in the calculated band structures (Supplementary Fig. S11), the top of the heavy-hole valence band (bottom of the conduction band) is above (below) the Fermi level as the Ho moments are tilted by 10 degrees from AFM state. On the other hand, the top of the valence band (bottom of the conduction band) shift underneath (over) the Fermi level as the Ho moments are completely aligned to the field, resulting in the disappearance of the hole and electron pockets. In terms of the Hall effect contribution, the Weyl points with opposite chirality are not exceptionally separated so that the consequent anomalous Hall effect is not significantly high and could be comparable the normal contribution. However, direct calculation of

the Hall data entails the estimation of carrier relaxation time in a first-principles way (i.e., simulating the scattering of electrons or holes with phonons, boundaries, and even disorders), which is extremely computationally expensive and inaccessible. To gain an insight into the Weyl points, other transport measurements such as planar Hall effect, which can detect the Berry curvature contribution without normal Hall effect (Kumar, N., Guin, et al., PRB 98, 041104(R) (2018)), may be a possible future work.

We have added sentences at P.9 L. 148-157 in the revised main text, and Supplementary Note III and Fig. S6 in the revised Supplementary Information.

Summary of revisions

Response to Referee #1

Comment 1: We have revised Fig. 2 and added the description of the sample dimension in Methods. We have also added the last subsection in **Supplementary Note I** and the sentences in **Supplementary Note IV**, and **Figs. S3** and **S8**.

Comment 2: We have revised the sentences at **P.9 L.165-169**. We have also added the explanations in **Supplementary Note VI** and **Fig. S12**.

Comment 3: we have added the sentences at **P.7 L.115-122** and **Refs. [24-28]**.

Comment 4: we have revised the sentences in the band calculation part in **Methods**.

Response to Referee #3

Comment 1: We have revised the sentences at **P.5 L.50-55** and **Methods** in the main text and added the **Refs. [3-4]**, and **Figs. S1e, S1f, and S3** in Supplementary Information.

Comment 2: We have added the sentences at **P.7 L.128-130** in the main text and added the explanations in **Supplementary Note IV** and **Fig. S8** in Supplementary Information.

Comment 3&4: We have revised the sentences at **P.12 L.201-205** in the main text and added the sentences in **Supplementary Note V** and **Figs. S9, S10, and S11**, and **Refs. [7-8]** in Supplementary Information.

Comment 5: We have added sentences at **P.9 L. 148-157** in the main text, and **Supplementary Note III** and **Fig. S6** in Supplementary Information.

REVIEWER COMMENTS

Reviewer #1 (Remarks to the Author):

Through the revision, the Authors have improved their manuscript considerably. However, two issues remain.

1. The explanation of a decrease of resistance by 4.5 orders of magnitude in the longitudinal configuration ($H \parallel I$) put forward in the manuscript is qualitative and, therefore, not truly conclusive. However, I would leave this interesting question to future works.
2. It still appears to me that a non-monotonic dependence of resistance on the magnetic field H in the transverse configuration is due to an admixture of ρ_{xy} and, accordingly, I claim that the values of $\rho_{xx}(H)$ were not properly determined. I would recommend comparing resistance measurements in the transverse orientation (H perpendicular to I) obtained for the case $l = w$, where l is the distance between the voltage probes and w the sample width (inset Fig. 2) to the case, say, $l = 3w$ [for a discussion of geometrical magnetoresistance, see, e.g., D. R. Baker and J. P. Heremans, Phys. Rev. B 59, 13927 (1999)]. Such a comparison would constitute a great service for the community, as the same question concerns, e.g., results presented in Ref. 12.
3. A side remark – is not a positive magnetoresistance in the inset to Fig. 2 caused by an effect of spin-splitting upon quantum localization corrections due to electron-electron interactions? (see, Ref. 28).

Reviewer #3 (Remarks to the Author):

I read the rebuttal letter and the revised manuscript "Colossal negative magnetoresistance in field-induced Weyl semimetal of magnetic half-Heusler compound". In general, I had the impression that

the authors tried to adequately respond to my comments and criticism. Overall, the paper is well written and presents experimental data which are at the cutting edge of what is currently possible. The data are thoroughly analyzed and most observations are reasonably supported either theoretically or by supplemental data.

However, I do not see sufficient general interest why the entire natural science community should care about these results. Although the HoAuSn shows a large negative magnetoresistance, the field-induced reconstructed band structure have been well studied in many half-Heusler alloys, such as in Refs. [5] and [6], and in fact, the chiral anomaly induced large negative magnetoresistance has been reported by Chen et al. in half-Heusler compounds $R\text{PtBi}$ ($R=\text{Tb, Ho, and Er}$) in *Applied Physics Letters*, 2020, 116(22): 222403. – see also references therein.

On the whole, many aspects have already been covered in above Refs and the data are not sufficiently unambiguous to support a much better understanding of the Weyl physics than what has been achieved previously. Nevertheless, I consider the results as highly interesting for a more focused readership, for example in a material-related Journal.

Authors' response to the Reviewer #1

Through the revision, the Authors have improved their manuscript considerably. However, two issues remain.

First of all, we thank the Reviewer for his/her kind review of the manuscript.

(1-1) comment

The explanation of a decrease of resistance by 4.5 orders of magnitude in the longitudinal configuration (H || I) put forward in the manuscript is qualitative and, therefore, not truly conclusive. However, I would leave this interesting question to future works.

(1-1) reply

We had a quantitative discussion about the field-induced semimetal state using the Fermi surface cross section, effective mass, and relaxation time extracted from the quantum oscillation (Fig. 3d and 3e), and based on these parameters, we performed the band calculation (Figs. 4, S9-11). Especially the latter was done rigorously in response to the previous reviewer's comments. We also pointed out the deviation from the relation $\frac{\rho_{xx}(M)}{\rho_{xx}(0)} = 1 - C \left(\frac{M}{M_S}\right)^2$ (Kondo lattice model) at low temperatures as shown in the inset of Fig. 2, so as to discuss the origin of the observed magnetoresistance in detail. To elucidate it, it would be promising to obtain an insight into the band structure by means of spectroscopy in the future. We have added a perspective at P.12 L.231-233.

(1-2) comment

It still appears to me that a non-monotonic dependence of resistance on the magnetic field H in the transverse configuration is due to an admixture of rho_xy and, accordingly, I claim that the values of rho_xx(H) were not properly determined. I would recommend comparing resistance measurements in the transverse orientation (H perpendicular to I) obtained for the case I = w, where I is the distance between the voltage probes and w the sample width (inset Fig. 2) to the case, say, I = 3w [for a discussion of geometrical magnetoresistance, see, e.g., D. R. Baker and J. P. Heremans, Phys. Rev. B 59, 13927 (1999)]. Such a

comparison would constitute a great service for the community, as the same question concerns, e.g., results presented in Ref. 12.

(addendum):

I am sorry to let you know that in my report to that paper I was wrong in point 2:

...

Actually for the resistivity tensor $[\rho_{xx}, \rho_{xy}; -\rho_{xy}, \rho_{xx}]$, the Maxwell equations for the stationary case: $\text{div } j = 0$ and $\text{rot } E = 0$ imply, for $\text{div } j = 0$, and spatially uniform sample, $\text{rot } j = 0$. This means, together with the condition $j_y=0$ at the x boundary (for current along x), that $\rho_{xx}(B=0)/\rho_{xx}(0)$ is not affected by the ratio of sample width W to length L . If L is not sufficiently greater than W , $\rho_{xx}(0)$ is slightly underestimated but this is not relevant in the present case, in which magnetoresistance is discussed. In this context, samples are spatially uniform, i.e., ρ_{ij} do not depend on the position, if regions of dimensions of the phase coherence length L_ϕ are identical, e.g., in terms of the impurities' number, and $L, W \gg L_\phi$.

Taking that into account I recommend the manuscript for publication, and apologize for my mistake.

(1-2) reply

We thank the Reviewer for the careful considerations on the measurement of resistivity and Hall resistivity. We agree that we should take into account the geometry of the electric-voltage measurements especially for materials with a very high carrier mobility. We think that the present data are deduced correctly since, in addition to the Reviewer's inquiry on the conductivity tensor, we carefully prepare the samples (Methods and the former part of Supplementary Note I) and check the sample dependence (the latter part of Supplementary Note I and Fig. S3).

(1-3) comment

A side remark – is not a positive magnetoresistance in the inset to Fig. 2 caused by an effect of spin-splitting upon quantum localization corrections due to electron-electron interactions? (see, Ref. 28).

(1-3) reply

We thank the referee for the instructive suggestion. We attribute the positive magnetoresistance at low fields below 20 K to the scattering on crystal-electric-field levels of Ho^{3+} $4f$ orbitals, since it is not observed in other rare-earth materials (not shown here). As discussed in P.5 L.61-67 and Ref. 17, the first excitation energy of the Ho^{3+} ground state multiplet in this system is about 20 K, at which the resistivity shows an anomaly (Fig. 1b). This is presumably because the fluctuation of $4f$ electrons between the splitting levels contributes to the increase of resistivity. In this sense, the positive magnetoresistance at low magnetic fields below 20 K may be due to the Zeeman splitting of the multiplet, which revives the fluctuation causing the electron scattering.

We have added the sentence at P.6 L.124-126 to explain the possible origin of positive magnetoresistance at the low magnetic field below 20 K.

Authors' response to the Reviewer #3

I read the rebuttal letter and the revised manuscript "Colossal negative magnetoresistance in field-induced Weyl semimetal of magnetic half-Heusler compound". In general, I had the impression that the authors tried to adequately respond to my comments and criticism. Overall, the paper is well written and presents experimental data which are at the cutting edge of what is currently possible. The data are thoroughly analyzed and most observations are reasonably supported either theoretically or by supplemental data.

We thank the Referee for constructive comments and approval for the revision of the manuscript.

However, I do not see sufficient general interest why the entire natural science community should care about these results. Although the HoAuSn shows a large negative magnetoresistance, the field-induced reconstructed band structure have been well studied in many half-Heusler alloys, such as in Refs. [5] and [6], and in fact, the chiral anomaly induced large negative magnetoresistance has been reported by Chen et al. in half-Heusler compounds RPtBi (R=Tb, Ho, and Er) in Applied Physics Letters, 2020, 116(22): 222403. – see also references therein.

On the whole, many aspects have already been covered in above Refs and the data are not sufficiently unambiguous to support a much better understanding of the Weyl physics than what has been achieved previously. Nevertheless, I consider the results as highly interesting for a more focused readership, for example in a material-related Journal.

We respectfully challenge to the Reviewer about the criticism on the novelty/significance of the present work, because the present findings are not trivial and thus can be hardly explained by previous theories and experiments. There are two important points as follows.

First of all, the single crystal of HoAuSn is newly synthesized. Back to more than a decade ago, motivated by the first demonstration of the quantum spin Hall effect in the HgTe/CdTe quantum well, two Nature Materials papers (Refs. 4 and 5)

suggested the half-Heusler alloys as promising candidates for various topological electronic states with high tunability. This stimulated abundant experimental studies. However, most of them were concentrated on the zero-gap semiconductor RPtBi , and the most important feature of this family of materials, i.e. tunability of the electronic states, has remained elusive so far. Therefore, the single crystal growth of the material, which is in the unexplored electronic phase, has a considerably high impact on this research field, since it expands the range of accessible electronic states and hence realizes the full potential of half-Heusler alloys, as highlighted especially in Fig. 1a.

In addition, we find the colossal magnetoresistance distinct from magnetotransport properties in conventional topological systems such as negative magnetoresistance induced by chiral anomaly. As the reviewer recognized after our reply, the exchange splitting indeed plays a key role in the magnetotransport properties in this series of materials. Nonetheless, what we stress here is that the observed negative magnetoresistance is several orders of magnitude larger than those reported in other topological materials, indicating that the exchange splitting can give rise to unprecedented phenomena in HoAuSn . Moreover, the field-induced semimetal state in this system shows the remarkably large residual resistivity ratio (RRR) of ~ 1000 and the small residual resistivity of $\sim 0.2 \mu\Omega\text{cm}$. This feature is unusual since many (half-)Heusler alloys are subject to crystalline defects. For instance, Co_2FeSi shows RRR of 5.2 and the residual resistivity of $1.5 \mu\Omega\text{cm}$ (see Ref. 19). The resistivity of rare-earth half-Heusler alloys reported so far is basically larger than $100 \mu\Omega\text{cm}$. Therefore, we believe that the present observations are highly exceptional in (half-)Heusler alloys and meet the criteria of Nature Communications.

Summary of revisions

Response to Referee #1

Comment 1: We have added the sentence at **P.12 L.231-233**.

Comment 3: we have added the sentence at **P.6 L.124-126**.